# Accelerate Scaling of LLM Finetuning via Quantifying the Coverage and Depth of Instruction Set

## Abstract

Scaling the amount of data used for supervied fine-tuning(SFT) does not guarantee the proportional gains in model performance, highlighting a critical need to understand what makes training samples effective. This work identifies two fundamental dataset properties that govern SFT scalability: **semantic coverage**, or the breadth of task domains, and **information depth**, or the richness of individual examples. We demonstrate that simple proxies for these properties explain the majority of validation loss variance in our experiments. In this work, we further propose the **Information Landscape Approximation (ILA)**, a model-agnostic data selection framework that jointly optimizes for these two factors. ILA constructs compact subsets that approximate the informational value of large datasets. Empirical results show that models tuned on ILA-selected data achieve faster and more sustained performance improvements across diverse tasks and model sizes compared to existing methods, a phenomenon we term **accelerated scaling**.

## 1 Introduction

Supervised fine-tuning(SFT) has emerged as a standard technique for adapting pretrained large language models(LLMs) to downstream tasks Zhang et al. (2023); Chung et al. (2022). However, empirical studies consistently reveal a scaling paradox: merely expanding the size of instruction-tuning datasets cannot guarantee the performance improvements Xia et al. (2024a); Zhang et al. (2024). This phenomenon reveals an important question: *What underlying properties of training data govern the scalability and efficiency of SFT?*

The core objective of SFT is not merely to memorize a set of examples, but to efficiently stimulate and re-organize the model's knowledge to follow instructions and solve tasks generalizably Zhou et al. (2023); Bai et al. (2022). The learning process is fundamentally constrained by the informational sufficiency of the training dataset Kaplan et al. (2020); Hoffmann et al. (2022). We assume that the efficacy of SFT scaling is governed by two axises of this informational sufficiency: **semantic coverage** and **information depth**.

Semantic coverage, on one hand, dictates the *breadth* of a dataset, answering the question: *Does the dataset expose the model to all necessary types of tasks?* It measures the diversity of semantic domains or task families(e.g., mathematics, summarization, coding) Wang et al. (2022); Bukharin and Zhao (2023). High coverage ensures that the fine-tuning process activates and adjusts the model's parameters across the full spectrum of capabilities required for generalization, preventing under-specialization in key areas Dong et al. (2023); Liang et al. (2025).

Information depth, on the other hand, dictates the *density* of a dataset, answering the question: *Does each training sample provide a substantial learning signal?* It quantifies the richness and complexity of task-relevant information within an individual example Li et al. (2023); Du et al. (2023). A sample requiring multi-step reasoning, integration of sub-skills, or understanding of nuanced concepts possesses greater depth than a simple, one-step query Yu et al.; Hendrycks et al.. Depth ensures that the model is not just exposed to a task type, but is compelled to engage in non-trivial pattern recognition and knowledge application Zhao et al. (2024); Achiam et al. (2023).

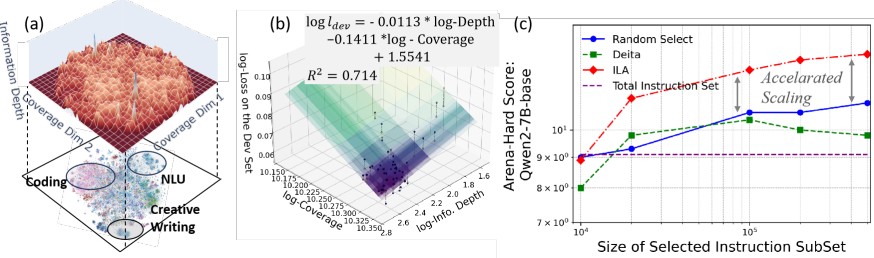

Figure 1: (a) Illustration of the information depth, coverage, and domain distribution of an instruction set; (b) The dev-loss of an finetuned model can be well fitted using the information depth and coverage of the instruction set for fine-tuning; (c) Performance of ILA scales up faster than simply enlarging the size of instruction set and SoTA instruction selection methods, suggesting a "*accelerated scale*" behavior.

We combine theoretical argument and empirical analysis to show the practical importance of these two axes. Using simple proxy measures for coverage and information depth, a linear regression explains a large fraction (over 70%) of validation-loss variation in our experiments (see Figure 1 (b)). This indicates that these two factors capture the dominant directions that determine SFT effectiveness under our setup.

Based on this insight, we develop the **I**nformation **L**andscape **A**pproximation (**ILA**) algorithm to refine SFT pools. ILA (i) provides reproducible proxy metrics for coverage and depth, (ii) selects subsets that approximate the information landscape of a large pool by jointly maximizing coverage and depth, and (iii) is intentionally simple and model-agnostic to encourage practical adoption Cao et al. (2023); Ge et al. (2024). Empirically, ILA-selected subsets improve model performance more quickly per added sample than random sampling and recent refinement baselines Xia et al. (2024b); Liu et al. (a) — an effect we term *accelerated scaling* (illustrated in Figure 1 (c) and Figure 4).

## 2 THE COVERAGE AND DEPTH OF AN INSTRUCTION SET DOMINATES THE PERFORMANCE OF THE FINETUNED MODEL

A key characteristic of the Supervised Fine-Tuning (SFT) process is that, at this stage, the pretrained model has already acquired substantial prior knowledge (Zhao et al., 2024). Therefore, unlike the pretraining stage where the total number of tokens can be used to measure the information within a dataset, in the SFT stage, what kind of *additional information* the instructions could bring in plays a critical role in determining the performance of the finetuned model and further governs the scaling regularity of the SFT process. Despite its crucial importance, due to the complexity of the instruction set distribution, previous work only modeled such effects using a constant Dataset Factor (Zhang et al., 2024). This restricts the practical guidance in constructing and refining instruction sets. In this section, our theoretical analysis shows that *coverage* and *information depth* are key factors within instruction distributions that influence model performance. After quantifying the coverage and information depth of an instruction set, experimental studies suggest a strong positive correlation between the model performance and the coverage and information depth of an instruction set.

### 2.1 THEORETICAL ANALYSIS

In this section, we show that in theory, the *coverage* and *information depth* of an instruction set could be key factors deciding the performance of a base model fine-tuned on it. Specifically, the supervised finetuning process aims at adapting pretrained LLMs to downstream tasks by finetuning them with instructions. Formally, the objective function could be characterized as:

$$L = \frac{1}{N} \sum_{i=1}^{N} CE(y_i|x_i), \tag{1}$$

where $x_i$ and $y_i$ is the query and response of an instruction $I_i$, respectively; $I = \{I_1, \ldots, I_N\}$ is an instruction set and $N$ is its size, $CE$ is the cross entropy loss. Essentially, an instruction could be

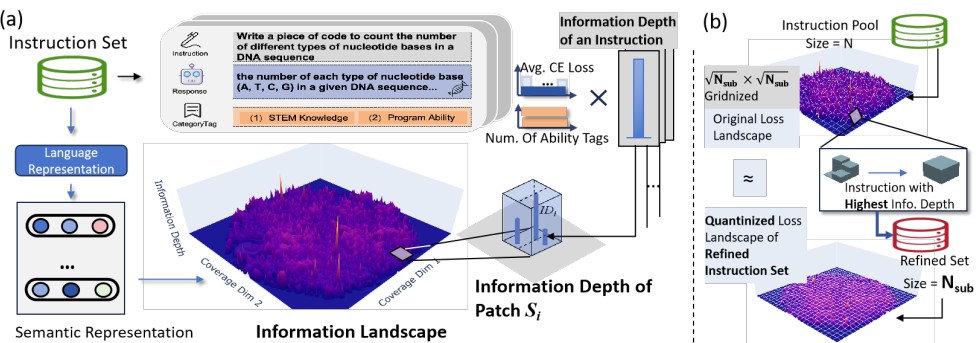

Figure 2: (a) The calculation of the proxy indicators measuring the information depth and coverage of an instruction set, which further forms into a landscape characterizing the distribution of an instruction set. (b) Illustration of the information landscape approximation (ILA) instruction refinement algorithm, which makes the information landscape of the selected subset approximate that of the original instruction pool.

regarded as a point in a semantic space $\mathcal{S} \in \mathbb{R}^d$. Thus, the above equation could be reformulated:

$$L = \frac{1}{N} \sum_{i=1}^{N} \sum_{j=1}^{d} l(z_{i,j}), \tag{2}$$

where $z_{i,j}$ represent the $j$-th dimension of the semantic representations of the $i$-th instruction, respectively, and $d$ is the dimensionality of the semantic space $\mathcal{S}$. Thus, the performance of the SFT is driven by: (1) the spatial distribution of instruction within the semantic space; and (2) at each point $z_i \in \mathcal{S}$, how much additional information is provided.

The above equation implicitly assumes that all instructions independently contribute to model performance. However, one prominent characteristic of LLM is the strong generalizability. Specifically, for a base model $\mathcal{M}_b$, after learning $I_i$, it can well generalize to a small area $\Delta\mathcal{S}_i$ within the semantic space centered around $z_i$. Formally, by SFT $\mathcal{M}_b$ using $I_i$, it leads to a decrease of the loss function $\delta_i = CE_{\text{base}}(y_i|x_i) - CE_{\text{SFT}}(y_i|x_i)$, then for other instructions within $\Delta\mathcal{S}_i$, the loss value on them could also be expected to decrease accordingly, i.e., $\mathbb{E}_{j=_i, I_j \in \Delta\mathcal{S}_i}(\delta_j - \delta_i) < \epsilon_j$, where $\mathbb{E}_{j=_i, I_j \in \Delta\mathcal{S}_i}(\delta_j - \delta_i) < \epsilon_i$ is the expectation loss decrease after learning $I_i$ upon instructions belong to $\Delta\mathcal{S}_i$ besides $I_i$, $\epsilon_j$ is a small term. Since these instructions locate in a small region $\Delta\mathcal{S}_i$, it could assume that their content are similar and $\epsilon_j \approx \forall I_i, I_j \in \Delta\mathcal{S}_i$. Therefore, $\mathbb{E}_{j=_i, I_j \in \Delta\mathcal{S}_i}(\delta_j - \delta_i) < \epsilon_i$.

This implies that for a base model $\mathcal{M}_b$, *given a set of instructions $\{I_j^{(i)}\}_{j=0}^{|S_i|} \in \Delta\mathcal{S}_i$, the "amount" of information $\mathcal{M}_b$ can derive from it primarily depends on the most informative sample*, i.e., the one with largest $\delta_i$. This phenomenon has been observed in several previous practical investigations (Zhao et al., 2024; Zhang et al., 2023), where the performance of the finetuned model is primarily driven by a small number of instructions, while the remaining instructions contribute little to the overall performance. This is because, on the one hand, if we choose only one instruction $I_k^{(i)}$ from $\{I_j^{(i)}\}$ for SFT $\mathcal{M}_b$, using the sample with the largest loss decrease to train $\mathcal{M}_b$ will maximize the expected loss reduction across the other samples. On the other hand, since these instructions share similar semantic content, training $\mathcal{M}_b$ with more than one such instruction would lead to redundancy and would not incorporate significantly more information. Therefore, within $\Delta\mathcal{S}_i$, the additional information that $\{I_j^{s_i}\}$ can provide is largely determined by the instruction with the maximum loss decrease. Formally, we define this as the *information depth* of patch $\Delta\mathcal{S}_i$ centered at point $z_i$, i.e.,

$$ID_{\Delta\mathcal{S}_i} = \max_{j, I_j^{(i)} \in \Delta S_i} \delta_j. \tag{3}$$

Similarly, $\delta_j$ could be defined as the information depth of instruction $I_j^{(i)}$, which we denote as $ID_j$. The above equation indicates that, the instruction with the maximum information depth decides the information depth upon a subspace of the semantic space. Such analysis and our analyses in the following section indicate that, the performance of LLM cannot simply be improved by incorporating

more instructions, once the maximum information depth is not improved. Hence, on the semantic space $\mathcal{S}$, the total additional information instruction set $I$ brings could be reformulated as:

$$L = \frac{1}{N_{\mathcal{S}}} \sum_{i=1} ID_{\Delta\mathcal{S}_i} \to \int_{\mathcal{S}} ID_{\mathcal{S}} \ d\mathcal{S}, \tag{4}$$

where $\mathcal{S}$ is a subset of a $d$-dimension semantic space within $\mathbb{R}^d$, describing the span of all possible instruction data. Note that, $\mathcal{S}$ is only a *subset* of $\mathbb{R}^d$, it may not distribute the whole $\mathbb{R}^d$. $ID_{\Delta\mathcal{S}_i} = 0$ if there is no instance upon $\Delta\mathcal{S}_i$. This equation describes in each area of $\mathcal{S}$ (i.e., *coverage*), how much additional information is provided by the instruction set $I$ (i.e., *information depth*). Intuitively, as shown in Figure 2 (a), the *coverage* and *information depth* of an instruction set $I$ forms into an "*information landscape*" across the semantic space, which is the key characteristic of $I$, as it describes on what domain, how much additional information are provided to the base model.

## 2.2 PROXY INDICATORS QUANTIFYING THE INFORMATION DEPTH OF AN INSTRUCTION AND THE COVERAGE OF AN INSTRUCTION SET

Directly measuring the coverage and information depth of instructions is rather difficult. In this paper, we propose two proxy indicators. In the following section, we show that these proxy indicators could be effective as it can explain a substantial proportion of the performance on the test set.

**The Proxy Indicator for the Depth of an Instruction** To estimate the information depth of an instruction, one intuitive way is to compare the cross-entropy value of a base model $\mathcal{M}_b$ and a fine-tuned model $\mathcal{M}_{SFT}$. However, the cross-entropy loss is associated with the response length, making it susceptible to verbosity. Additionally, a single query $x_i$ may yield multiple valid responses of significantly divergent lengths. As a result, the estimation of information depth can be substantially confounded by response length. To address this issue, we notice that, the requisite skills or knowledge for addressing a query remain largely consistent. For instance, in a QA task, whether the response is succinct or not, the essential knowledge required—comprising factual information or reasoning capabilities, remains unchanged. Therefore, the additional information brought by an instruction should relate to the inherent number of skills or knowledge it encapsulates, rather than the response's verbosity. Hence, based on the cross-entropy loss, as illustrated in Figure 2 (a), to estimate the Information Depth, we first normalize the cross-entropy loss of instructions by dividing it with the response length and then multiply the avg-cross-entropy loss by the number of requisite skills or knowledge:

$$\widehat{ID}_j = \delta_j / T_j \times \#\text{label}, \tag{5}$$

where $T_j$ is the number of tokens within $y_j$. Several opensource projects provide tagging systems to obtain the ability or knowledge labels (Lu et al., 2023; Zhao et al., 2025). In this paper, we adopt the method of Zhao et al. (2025).

**The Proxy Indicator for the Coverage of an Instruction Set** To get the coverage span $\mathcal{S}_I$ of an instruction set $I$, we first project each instruction $I_i = (x, y_i)$ using a textual representation model. Henceforth, by uniformly cutting the whole semantic space into $g^d$ ($d$ is the dimension of $\mathcal{S}$) grids and calculating the number of grids with more than one instruction (denoted as $\widehat{\mathcal{S}}_I$), the coverage of an instruction set can be roughly estimated.

**Disentangling the Information Depth with the Coverage of an Instruction Set** Another issue is the correlation between the information depth and coverage. Specifically, in different regions of the semantic space, the values of the information depth vary, as the cross entropy loss varies. For example, the Cross-Entropy loss on math- and code-related instructions is generally lower than that on creative generation tasks, leading to a generally lower information depth. This correlation complicates obtaining instruction sets while independently controlling the coverage or information depth with the other factor changing. For instance, deriving subsets with high information depth would naturally lead to selecting more creative generation-related instructions.

To normalize such a confounding effect, we shift from using the absolute value of $\widehat{ID}_j$ to *relative information depth*. Specifically, given a patch $\Delta\mathcal{S}_i$ and a set of instructions $\{I_j^{(i)}\} \in \Delta\mathcal{S}_i$, instead of using the information depth $\widehat{ID}_j$, we derive the relative information depth by calculating the

quantile of $I_j$ for $I_j \in \{I_j^{(i)}\}$. We denote the relative information depth of $I_j$ as $\widehat{RID}_j$. Formally, $\widehat{RID}_j = 1 - q(I_j)$, where $q(\cdot)$ is the quantile function. $\widehat{RID}_j$ is comparable across domains; for instance, the instructions with top 1% highest information depth (i.e., RID=0.99) in the math domain are deemed to have a higher quantile than the 50% quantile (i.e., $\widehat{RID}$=0.5) in the creative writing domain, yet in absolute terms, they may be lower. In this way, the correlation between the information depth and coverage of the instruction set is disentangled, and we can independently investigate the influence of coverage or information depth.

### 2.3 EMPIRICAL SCALING REGULARITY BETWEEN MODEL PERFORMANCE WITH DEPTH AND COVERAGE OF AN INSTRUCTION SET

To investigate the scaling regularity between the performance of a finetuned model and the depth and coverage of an instruction set, we construct a series of instruction sets with varying coverage and information depth, then finetune a base model on these instruction sets to observe how model performance changes with the coverage and depth of instruction sets. Specifically, we: (1) Control the size and coverage of instruction sets, while varying the information depth; (2) Control the size and information depth of instruction sets, while varying the coverage. Empirical analyses show that, using these two proxy indicators, the performance on the test set could be largely explained.

To this end, we first collect a sufficiently large instruction pool $I$ and obtain the spatial distribution of instructions within the semantic space, along with estimating the information depth of each instruction. To draw instruction sets with different coverage and information depth, we: (1) Segment $\mathcal{S}_I$ into a set of patches $\{\Delta S_i\}$; (2) Calculate the frequency of instructions in each patch $\{\Delta \mathcal{S}_i\}$ and rank the patches based on frequency from high to low, we could obtain a sequence $\{\Delta^* \mathcal{S}_1, \ldots, \Delta^* \mathcal{S}_{|S|}\}$; (3) Assume $n_1 < \ldots n_l < \ldots n_L$, by merging the top $n_l$ patches $\{\Delta^* \mathcal{S}_1, \cdots, \Delta^* \mathcal{S}_{n_l}\}$, we can obtain a set of sub-regions $\mathcal{R}_l, \ldots, \mathcal{R}_l, \ldots, \mathcal{R}_{n_L}$, where each sub-region is a union of patches and $\mathcal{R}_l$ is a true subset of $\mathcal{R}_{l+1}$. In other words, a set of sub-regions with low to high coverage can be obtained. For each sub-region $\mathcal{R}_l$, we can select $N_{sub}/n_l$ instructions within each patch in $\mathcal{R}_l$, with the relative information depth to $RID < \tau$. In this manner, we can select subsets of instructions with fixed size $N_s ub$, fixed information depth $\tau$, and varying coverage $\mathcal{R}_l, \ldots, \mathcal{R}_l, \ldots, \mathcal{R}_{n_L}$. Similarly, given a sub-region $\mathcal{R}_l$, by selecting $N_{sub}/n_l$ instructions from the $RID < \tau_1, \ldots, \tau_t$ regions, we could obtain a set of subsets with fixed size $N_{sub}$ and fixed coverage $\mathcal{R}_l$, while information depth varies from $\tau_1$ to $\tau_t$.

Taking the coverage and depth of instruction sets as dependency variables, and the performance on the development set as the dependent variable, we could fit a linear regression function. Formally:

$$\log L_{dev}^l = \beta_0 + \beta_1 \log \widehat{RID}_l + \beta_2 \log \mathcal{S}_{\mathcal{R}_l}, \tag{6}$$

where $\beta_k$ is regression coefficients, $L_{dev}^l$ is the mean cross entropy loss value on the development set of model finetuned upon the $l$th instruction set, $\mathcal{S}_{\mathcal{R}_l}$ is the coverage area of sub-region $\mathcal{R}_l$ within the semantic space.

### 2.3.1 EXPERIMENTAL SETTINGS

We employ InfinityAtlas as the whole instruction pool (Zhao et al., 2025), which is a large-scale scale high-quality instruction collection containing 2 million high quality instructions with large enough coverage, and hard enough instructions, **together with a set of labels describing the necessary skills or knowledge for completing one instruction**. To evaluate the performance of the finetuned model, we randomly sampled 20% of instructions to obtain a development set, and left all the other as the instruction pool $I$ for selecting subsets.

To obtain the spatial distribution of the instructions, we get the representation vectors using BGE (Xiao et al., 2024). Then we use t-SNE (Van der Maaten and Hinton, 2008) to reduce the representation vectors into a 2-dimensional plane to alleviate the sparsity of instruction within the semantic space caused by the high-dimensionality. Henceforth, by uniformly cutting the whole semantic space into $g \times g$ grids and calculating the number of grids with more than one instruction, the coverage of an instruction set can be roughly estimated. For a grid $g$ with several instructions within $g$, for an arbitrary instruction $I_i \in g$, using a base model $M_b$ and a SFT-ed model $M_{\text{SFT}}$, we could derive the loss decrease $\delta_i = l_b(y_i) - l_{\text{SFT}}(y_i)$, and then obtain the information depth of $g$ given $I$.

To estimate the information depth of instructions, we use Llama3-8B (Dubey et al.) as the base model. To obtain $M_{\text{SFT}}$, we randomly sample a small subset $I_{\text{SFT}}$ from $I$, and employ $I_{\text{SFT}}$ to SFT a base LLM $M_b$ to obtain $M_{\text{SFT}}$, so that for an instruction $I_i \in \{I \setminus I_{\text{SFT}}\}$, by comparing the loss value obtained by $M_b$ and $M_{\text{SFT}}$, the loss decrease $\delta_i$ can be derived for estimating the information depth. Thus with the information depth of each instruction, from the instructions within the top $q$ quantile information depth, we could draw a subset $I_q^t$ with size $N_{sub} = 20k$, and calculate its coverage by arbitrarily cutting the semantic space into $500 \times 500$ grids and calculate the number of grids with more than one instruction as the coverage of instruction set. So that we could obtain instruction sets with different depths and coverage. In practice, a total of 36 datasets are drawn. More details are provided in the Appendix D.

### 2.3.2 ANALYSIS RESULTS

Figure 3 shows the result of regression analysis on instruction sets with different coverage and depth. From which we have the following observations: (1) The regression coefficient over the information depth and coverage of instruction is highly statistically significant and negative, suggesting that **the performance of the finetuned model is strongly positively correlated with both the width and coverage of an instruction set**, i.e., high performance corresponding to lower dev-loss and higher information depth and coverage.

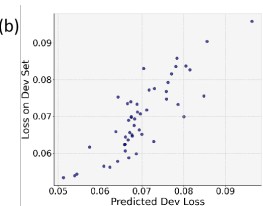

Moreover, these two independent variables can account for over 70% of the variance in the performance loss of the LLM on the development set, representing a dominant proportion. This indicates that the effects of an instruction set with a relatively complex distribution can be explained by a rather limited number of key factors, showing the rationality of our theoretical analysis and the effectiveness of two proxy indicators. This would further provide for instruction set optimization methods based on directly quantifying the information depth and coverage of an instruction set.

Figure 3: (a) Regression results of the dev-loss vs. coverage and depth of instruction sets; (b) Scatter plot of predicted vs. actual dev loss.

(2) For comparison, instead of using $\widehat{RID}_j$ as the proxy indicator of information depth, we also conduct experiments with taking $\delta_j$ as the metric of information depth. Results are provided in the Appendix. Briefly, $\delta_j$ is significantly less predictive compared to $\widehat{RID}_j$. This suggests the necessity of accounting for the verbosity and style of responses when estimating the information depth of instructions, and the reasonability of our proposed proxy indicator.

(3) Several error sources may exist, including: i) textual representation models may not be able to accurately project the instructions into semantic space, and ii). The potential impact of the t-SNE dimensionality reduction process on calculating the accuracy of spatial coverage. Nevertheless, a substantial proportion of the performance could be explained with the existence of such potential error sources, suggesting the effectiveness of these two proxy indicators.

## 3 ACCELERATE SCALING BY OPTIMIZING THE COVERAGE AND DEPTH OF INSTRUCTION SET

### 3.1 METHODOLOGY

Since the additional information an instruction set could bring to a base model is largely characterized by its information landscape, if we could select a subset from the pool, with coverage and depth as similar as that of the original pool, then it could be possible to accelerate approaching the *information landscape* of an instruction pool compared to simply incorporating more instruction set (i.e., "*SuperScale*"). To this end, as shown in Figure 2 (b), we devise a **I**formation **L**andscape **A**pproximation (**ILA**) algorithm.

Heuristically, refining the instruction set aims to select a subset $I_{sub}$ that could bring additional information to a base LLM similar to that of the original instruction pool $I_{ori}$. Note that the information landscape characterizes the additional information of the original instruction pool. Hence, given $I_{ori}$ with size $N_{ori}$ and is gridded into patches with size $\mathcal{S}$, to select a subset with size $N_{sub}$, the goal of refinement could be to select a subset with similar information landscape, which could be achieved by gridding the information landscape of $I_{ori}$ in $d-$dimension into $(N_{ori}/N_{sub}) \cdot S$ resolution. To this end, as described in Figure 2 (b), given an instruction pool $I_{ori}$, we first project each instruction into a $d$-dimensional semantic space and obtain the information depth of each instruction. Then to select a subset of size $N_{sub}$, we uniformly cut the coverage of $I_{ori}$ into $N_{sub}$ patches, with $N_{sub}^{1/d}$ segments in each dimension. Then within each patch, the instruction with the maximum information depth is selected into $I_{sub}$. In this way, the coverage of $I_{ori}$ is kept (i.e., coverage first), meanwhile with local information depth maximized. Moreover, heuristically, multiple instructions with different information depths. Therefore, in these regions, instructions with lower information depth may be redundant and should be excluded from the refined instruction set. By making the coverage of the refined instruction set close to that of the original instruction set and keep the coverage, the information density of the instruction set can be enhanced, thereby improving the performance of the corresponding model.

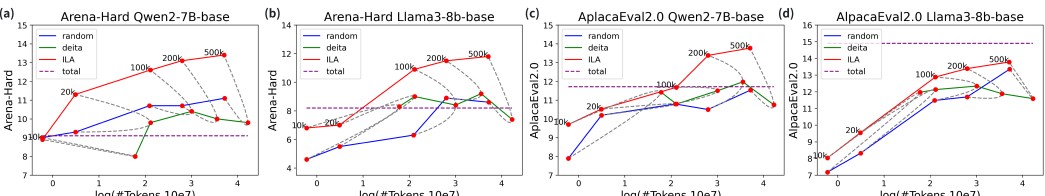

Figure 4: The x-axis represents the number of tokens, the y-axis shows the evaluation metric scores; the dashed lines connect results obtained using an equal number of instructions.

## 3.2 EXPERIMENTAL ANALYSIS

We conduct experiments on the general domain instructions and a reasoning-intensive math-solving task to evaluate the effectiveness of the ILA strategy, and further examine the reasonability of our observations on the relationship between the coverage and depth of instructions with the performance of the finetuned model.

**Experimental Settings**  In practice, we find that simply representing $I_{ori}$ using a 2-dimensional space could achieve a satisfying performance. We also adapt InfinityAtlas as the instruction pool, and draw a series of sub-instruction sets with sizes of 10k, 20k, 100k, 200k, and 500k from it, using our proposed ILA algorithm, together with random selection, and SoTA instruction set refinement algorithm Deita (Liu et al., b) as baselines, which uses heuristic indicators to measure the *complexity* and *diversity* of an instruction set rather than directly measure the *information depth* and *coverage*. Then we finetune opensource base models Qwen2-7B-base (Chu et al., 2024) and LLaMA3-8B-base (Dubey et al.) on these sub-instruction sets, and evaluate the performance of these finetuned models using widely adopted benchmarks AlpacaEval 2.0 (Dubois et al., 2024) and ArenaHard (Li et al., 2024). Moreover, the performance of two base models fine-tuned on the whole instruction pool is also provided. More details are provided in Appendix B and Appendix D.

**Results and Analysis**  Figure 4 shows the model performance finetuned on subsets selected by Random Selection, Deita, and ILA, respectively. For comparison, we set the x-axis as the **total tokens in the response**, and link instruction sets with the same number of instructions using a gray dashed line. Moreover, the performance of finetuning the base models using **ALL** instructions is marked as "total", which represents the performance with all information within an instruction set exposed to a base model. From Figure 4 we observe that:

(1) As the size of the selected subset increases to 500k, the performance of Random Selection on both benchmarks continuously scales up. In contrast, Deita struggles in *scale up*: as the size of the selected subset increases, the benefits compared to Random Selection degrade, or even turn negative. This phenomenon is also observed in other heuristic indicator-based instruction selection methods

(Xia et al., 2024a). In contrast, upon both benchmarks, our proposed ILA consistently outperforms Deita and the random selection strategy with the same number of instructions and tokens, even when the size of the total instruction pool reaching 2 million, and the size of the selected subset reaches 500k, indicates a **superscaling** behavior that the performance could be continuously improved over simply incorporating more instructions. This demonstrates the effectiveness of our approach ILA in refining instruction sets. As ILA is built upon the theoretical analyses and the two proxy indicators, the advantage if ILA is that it provides empirical supports for the rationality of the theoretical analyses and two proxy indicators.

(2) A key observation is *adding more instructions doesn't always improve performance*. For example, as shown in Figure 4 (a) and (b), on the ArenaHard benchmark focusing on complex tasks, models finetuned on the full instruction set may perform worse than those trained on smaller subsets. This is likely due to redundancy in the instruction set, where instructions with different information depth coexist within the same semantic space, with low-information-depth instructions occupying a significant portion. This underscores the necessity of refining the instruction set.

### 3.3 ACCELERATED SCALING THROUGH ENHANCING THE COVERAGE AND INFORMATION DEPTH

To further investigate the resources of the performance improvement of our approach, we examine the value of the information depth indicator and coverage indicator of the subset selected by our approach, together with baselines Random Selection and Deita. As Figure 5 shows:

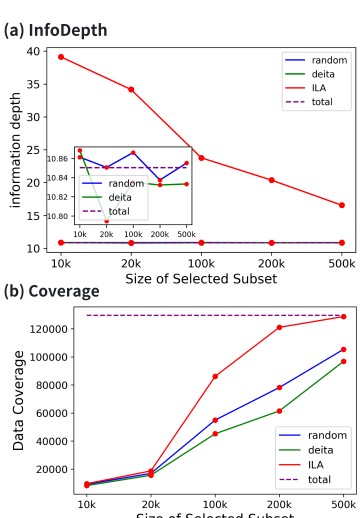

Figure 5: Information depth and coverage of subsets selected by ILA, Deita, and Random Selection.

(1) As the size of subsets increases from 10k to 500K, the information depth of instructions selected by Random Selection remains nearly invariant, while the coverage increases, together with the scale-up of model performance as shown in Figure 4. This demonstrates that, by simply randomly incorporating more instructions, the performances are mainly improved by expanding the coverage of instruction sets. However, it also implies inefficiency in enhancing the model performance, as the Random Selection process always tends to choose instructions in high-density regions, slowing the expansion of coverage, and failing to selectively incorporate more informative instructions.

(2) The heuristic indicator-based method Deita shows lower coverage compared to Random Selection, while has not improved the information depth of instructions. This could explain the performance priority of Deita when more than 100k instructions are included in the instruction set. In contrast, the instructions selected by ILA show consistently higher information depth and coverage compared to Random Selection and Deita, as well as performance advantages compared to baseline methods. These suggest that, by improving the coverage and depth of selected instruction sets, the performance of the finetuned model could be scaled up more efficiently, in turn supporting our theoretical analyses. Note that, as shown by Xia et al. (2024a), Deita could stand as a representative for a series of widely adopted instruction refinement methods that largely suffer from the inscalability of performance. We choose the SoTA Deita for comparison in this paper. This highlights the necessity of measuring the information depth and coverage of instruction sets.

(3) We also conducted experiments on base models with different sizes and obtained similar observations. Appendix B provides details.

### 3.4 VERTICAL DOMAIN EXPERIMENT

To further validate the effectiveness of our approach, we conducted experiments in math reasoning instructions, which require intensive reasoning ability, while with a restricted horizon compared to the open-domain instructions.

**Experimental Settings**   We aggregate four publicly available math-related instruction sets: Meta-Math (Yu et al.), QwQ-LongCoT-130K, QwQ-LongCoT-130K-2, and QwQ-LongCoT-Verified-130K. After rigorous clean and deduplication, a total of $\sim 650,000$ are left. Considering the inscalability of Deita, in this section, only Random Selection is used as a baseline. From the whole math-related instruction pool, three subsets with a size of 20k, 50k, and 100k are selected, then Qwen-Math-7B (Chu et al., 2024) is finetuned upon these subsets, and the test set of the MATH (Hendrycks et al.) dataset is employed as the benchmark to evaluate the performance of finetuned models. More details are provided in Appendix C.

| Method | 20k | 50k | 100k |
|---|---|---|---|
| Random choice | 0.5638 | 0.5914 | 0.5864 |
| **ILA (ours)** | **0.6224** | **0.6356** | **0.6492** |
| Absolute gain | 0.0586 | 0.0442 | 0.0628 |
| Relative gain (%) | +10.4% | +7.5% | +10.7% |

Table 1: Comparison between random-choice and ILA-selected subsets (accu. on the MATH dataset).

**Results and Analyses**   From Table 1, we observe that, on the MATH dataset, ILA consistently outperforms random selection as the size of the selected subsets increases. Heuristically, the model's performance on reasoning-insensitive tasks is more dependent on the depth of instructions. Hence, these results show the effectiveness of our approach in reasoning-intensive domains by effectively identifying instructions with high information depth. Moreover, as the size of the selected subsets increases, the performance of random selection either stagnates or declines. This indicates that simply incorporating more instructions may not necessarily lead to sustainable performance improvements (Xia et al., 2024b). Previous studies show that including low-information instructions can lead to performance degradation (Zhou et al., 2023), highlighting the necessity of refining the instructions to remove instructions with low information depth on the other hand.

## 4   RELATED WORK

**Scaling Laws for Finetuning**   The scaling law becomes more nuanced in the SFT stage compared to the pretraining stage, as the pretrained base already possesses substantial knowledge (Alba et al., 2025; Zhang et al., 2025). Previous analyses suggest that model performance after SFT is positively correlated with factors such as the size of the instruction set, the number of tasks within the instruction set, and the complexity of individual responses (Qin et al.). In this paper, we directly measure how additional information in coverage is provided, and the relationship between the performance of a finetuned model.

**Refinement of Instruction Set**   To derive subsets with a smaller size while models fine-tuned on them can achieve comparable or even better performance, main previous work selects informative instructions using heuristic indicators about the quality, complexity, and diversity of instructions (Wang et al., 2024; Ding et al., 2023; Chung et al., 2022; Shahzad et al., 2025). However, emerging evidence suggests that these methods struggle in *scale up*: as either the size of the whole instruction pool or the size of the selected subset increases, the benefits of these methods degrade, or even turn negative compared to just randomly selecting (Xia et al., 2024a). This significantly limits the practical application of these methods. Essentially, the refinement of the instruction set depends on the illustration of the scaling regularity between model performance and instruction distributions. By analyzing such a relationship, we propose an Information Landscape Approximation algorithm. Experimental results show that ILA selects instruction subsets with better performance than SoTA baselines and scales effectively, even as the full instruction pool grows to 3 million, and the subset reaches 0.5 million.

## 5   CONCLUSION

In this paper, we investigate the scaling behavior of LLMs in SFT and find that the coverage and information depth of an instruction set are significantly related to the performance of a base model fine-tuned on it. Based on such observation, we propose an **I**formation **L**andscape **A**pproximation algorithm to simultaneously maximize the depth and coverage of a refined instruction set. Experimental results demonstrate that ILA outperforms baseline methods, enabling more efficient and sustainable scaling of the finetuning process.

ETHICS STATEMENT

We confirm that our work has been conducted in accordance with the ICLR Code of Ethics (https://iclr.cc/public/CodeOfEthics). The study does not involve human subjects, sensitive personal data, or experiments that may cause harm to individuals or groups. All datasets used are publicly available and contain no personally identifiable information. Our methodology and findings are intended solely for academic research and do not present foreseeable risks of misuse. We have carefully considered potential concerns related to fairness, bias, and privacy, and our research adheres to recognized ethical standards.

REPRODUCIBILITY STATEMENT

We ensure reproducibility by providing an anonymized project repository on GitHub (https://github.com/Scaling-regularity-guided-instruction-synthesize.git). The construction of the instruction data pool is detailed in Appendix C, the construction of the evaluation indicators is detailed in Appendix D, the hyperparameter settings and experimental settings are detailed in Appendix E, and the mathematical formula experiments are detailed in Appendix F.

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

## A    USE OF LARGE LANGUAGE MODELS

We employed LLMs exclusively as editing assistants to enhance grammar, clarity, and conciseness of the manuscript. All technical contributions, including experimental design, data processing, evaluation, and conclusions, were conceived, implemented, and validated by the human authors. Edits suggested by LLMs were carefully reviewed and either accepted or modified by the authors; no numerical results, figures, or analyses were generated or approved solely by the LLM.

## B    MODEL SIZE SCALING EXPERIMENT

To further validate the universality and robustness of our approach, we conduct experiments across multiple model scales to examine whether the scaling behavior persists under different model capacities. These experiments aim to demonstrate that our method provides a general mechanism for quantifying the relationship between instruction data and model performance, rather than relying on specific model size or capacity.

**Experimental Settings**    We choose three representative models from the Qwen2 family with different parameter sizes: Qwen2-1.5B, Qwen2.5-3B, and Qwen2-7B (Chu et al., 2024). For each model, we apply our proposed quantification mechanism to select instruction subsets of size 10k, 20k, and 50k from a common instruction pool. Each subset is then used to fine-tune the corresponding model using standard SFT procedures. We adopt AlpacaEval 2.0 as the evaluation benchmark. More implementation details can be found in Appendix B.

| Model | 10k | 20k | 50k |
|---|---|---|---|
| Qwen2-1.5B | 2.72 | 3.08 | 4.35 |
| Qwen2.5-3B | 6.57 | 7.75 | 8.02 |
| Qwen2-7B | 11.68 | 13.38 | 13.77 |

Table 2: AlpacaEval 2.0 scores of our method across different model sizes and instruction scales.

**Results and Analyses**    As shown in Table 2, our method consistently produces increasing performance with larger instruction subsets across all three model sizes. This confirms that our quantification-based selection process is effective regardless of model scale. For smaller models such as Qwen2-1.5B, performance still steadily improves with data size, suggesting that even in low-capacity scenarios, identifying instruction subsets with high information value is beneficial.

The stability of the scaling trend across varying model sizes suggests that our method captures an intrinsic relationship between instruction data and model capability. This supports our central hypothesis: by modeling the information interaction between instruction data and the model, we can generalize a scalable instruction tuning framework that is both **model-agnostic** and **robust across capacity regimes**. Unlike prior approaches that rely on heuristic filters or manual data curation, our method offers a principled, automated perspective for quantitatively analyzing how instruction characteristics affect model learning. We believe this provides a new research direction for understanding and formalizing the role of instruction data in large-scale model training.

## C    INSTRUCTION POOL

To ensure comprehensive coverage of the main instruction categories, we first collect a sufficiently large set of instructions. Based on this collection, we exclude instructions that are not manually annotated or generated by advanced LLMs such as GPT-4 or ChatGPT. Additionally, we incorporate datasets such as Logi-QA, Wild-Chat, and COIG-CQIA. A detailed list of the included instruction set is provided in Table 3.

To mitigate duplicates, we apply SimHash with a threshold of 0.95. After this duplication removal process, and to ensure experimental stability, we retain only English-language instructions. The final instruction pool contains 1,994,253 instances.

| | |
|---|---|
| Alpaca GPT4 | LIMA |
| Alpaca GPT4 ZH | LongForm |
| BaiZe | logi-COT |
| BELLE Generated Chat | ShareGPT-Chinese-English-90k |
| BELLE Multiturn Chat | UltraChat |
| BELLE train 3.5M CN | Wizard Evol instruct zh |
| databricks-dolly-15K | Wizard Evol instruct 196K |
| BELLE School Math | Code Alpaca 20K |
| MetaMath | WildChat |
| COIG-CQIA | |

Table 3: List of instructions included for analysis.

## D    EVALUATION METRICS

We employ the following evaluation metrics to assess the performance of the finetuned large language models:

**AlpacaEval** (Length-Controlled AlpacaEval: A Simple Way to Debias Automatic Evaluators) is an evaluation framework designed to mitigate the impact of length biases in automatic evaluators. This metric is used to evaluate models on various tasks by controlling the length of responses, ensuring that the performance is not skewed by the length of the output.

**Arena-Hard** (From Crowdsourced Data to High-Quality Benchmarks: Arena-Hard and Bench-Builder Pipeline) is a dataset that emphasizes high-quality, crowdsourced benchmarks. Arena-Hard focuses on tasks that are particularly challenging for language models, providing a robust evaluation of model performance across a wide range of domains. We use Arena-Hard to assess the models on more complex, real-world problem sets.

**MATH** (Hendrycks et al., 2021) is a dataset consisting of challenging high-school math problems, categorized into the following topics: Prealgebra, Algebra, Number Theory, Counting and Probability, Geometry, Intermediate Algebra, and Precalculus. The problems in MATH are more difficult and diverse compared to those in GSM8K. In this paper, we use the open-source GitHub repository `gsm8k-ScRel` to evaluate the MATH scores. We also use 500 test problems from Lightman et al. (2023) as an out-of-domain math benchmark.

Each of these metrics provides a different perspective on model performance, ensuring a comprehensive evaluation of the finetuned large language models.

| Variable | Range |
|---|---|
| Label Length | $\{(1, 2, 3, 4), (5, 6), (7, 8), (9, 10), (11, 12), (13, 14), \geq 15\}$ |
| Label Frequency | $\{(0, 9], (9, 17], (17, 31], (31, 66], (66, 132], (132, 503], > 503\}$ |
| Base Loss | $\{(0, 0.713], (0.713, 0.948], (0.948, 1.125], (1.125, 1.369],$ $(1.369, 1.576], (1.576, 2.02], > 2.02\}$ |

Table 4: Ranges of label length, label frequency, and base loss value used in analysis.

## E    HYPERPARAMETER SETTINGS FOR SFT

We fine-tune all SFT datasets for 3 epochs with a batch size of 128 using NVIDIA H100 GPUs. For the 7B and 8B models, we utilize 8 GPUs. The learning rate is set to $9.65 \times 10^{-6}$, and the learning rate follows a cosine decay schedule. We evaluate the results at the final epoch.

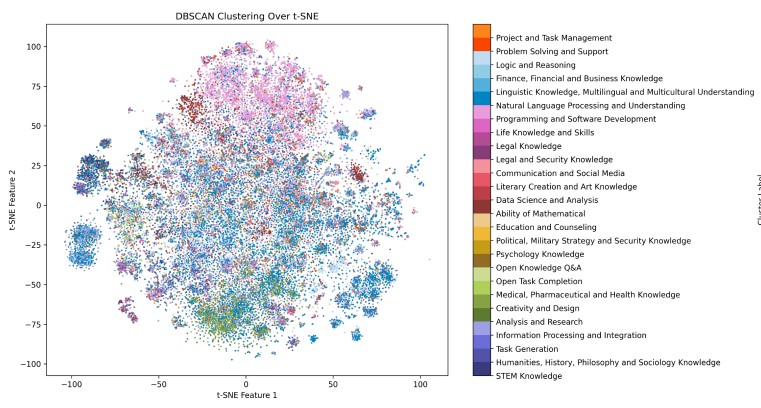

Figure 6: Spatial Distribution of instructions with different ability labels within the semantic space.

# F MATHEMATICAL FORMULA EXPERIMENT

EXPERIMENTAL DESCRIPTION

This experiment investigates the impact of different data selection criteria on the **loss of a supervised fine-tuning (SFT) model**. We focus on three key data selection variables:

- **X1 (Base Loss)**: The loss value of the original data on the base model, indicating data difficulty.
- **X2 (Label Frequency)**: The frequency of label occurrences in the original dataset, representing data representativeness.
- **X3 (Label Length)**: The length of the label list, reflecting the semantic richness of the data.

In our experimental design, we first divide the data into **seven equal partitions** based on X1, X2, and X3 (each with a uniform proportion of 1) and compute the **average loss (Y1)** of the SFT model. Then, we systematically adjust the proportion of a specific partition (e.g., increasing one partition to 2 or 3 while keeping the others at 1) to observe the changes in Y1. This allows us to analyze the correlation between Y1 and the three selection variables. The experimental results, as shown in Table 2, provide insights into how different data selection strategies influence the performance of the SFT model.

# G VISUALIZATION OF TEXTUAL SEMANTIC CLUSTERS USING t-SNE AND DBSCAN

## G.1 EXPERIMENT OVERVIEW

To analyze the semantic distribution of text data within our dataset, we employ **t-SNE (t-Distributed Stochastic Neighbor Embedding)** for dimensionality reduction and **DBSCAN (Density-Based Spatial Clustering of Applications with Noise)** for clustering. This visualization provides insights into the semantic structure and categorical distribution of textual data.

## G.2 METHODOLOGY

1. **Dimensionality Reduction**: We use t-SNE to map high-dimensional text embeddings into a 2D space, preserving local similarities. 2. **Clustering Algorithm**: DBSCAN is applied to identify dense clusters of semantically similar texts while marking noise points. 3. **Color Encoding**: Each category in the dataset is assigned a unique color, as shown in the legend, to represent different text classes.

| experimental Group | Experiment Name | Y1 | X1 | X2 | X3 |
|---|---|---|---|---|---|
| Base Loss (7 parts) | Base Loss-1 | 1.0785 | 0.558 | 183.7902 | 8.5869 |
| | Base Loss-2 | 1.0896 | 0.7007 | 185.1827 | 8.8201 |
| | Base Loss-3 | 1.0807 | 0.8122 | 168.3147 | 9.052 |
| | Base Loss-4 | 1.0803 | 0.9184 | 160.3745 | 9.1376 |
| | Base Loss-5 | 1.0866 | 1.0273 | 158.5329 | 9.1114 |
| | Base Loss-6 | 1.0617 | 1.1499 | 169.5216 | 9.0078 |
| | Base Loss-7 | 1.0714 | 1.3933 | 173.9337 | 8.6918 |
| Base Loss (7 parts, 1 with ratio 2) | Base Loss-2-1 | 1.0672 | 1.291 | 183.7112 | 8.6668 |
| | Base Loss-2-2 | 1.0718 | 1.3241 | 180.2553 | 8.7342 |
| | Base Loss-2-3 | 1.0643 | 1.3501 | 174.3433 | 8.7996 |
| | Base Loss-2-4 | 1.0768 | 1.3752 | 179.2039 | 8.7762 |
| | Base Loss-2-5 | 1.0605 | 1.4039 | 178.8521 | 8.7392 |
| | Base Loss-2-6 | 1.0665 | 1.4408 | 187.0676 | 8.6688 |
| | Base Loss-2-7 | 1.0733 | 1.5788 | 195.5053 | 8.4541 |
| Base Loss (7 parts, 1 with ratio 3) | Base Loss-3-1 | 1.0548 | 1.2083 | 180.3942 | 8.6839 |
| | Base Loss-3-2 | 1.0554 | 1.2715 | 172.2513 | 8.7812 |
| | Base Loss-3-3 | 1.0558 | 1.3144 | 169.869 | 8.8606 |
| | Base Loss-3-4 | 1.0636 | 1.3585 | 173.9858 | 8.8406 |
| | Base Loss-3-5 | 1.0581 | 1.4092 | 176.9452 | 8.7716 |
| | Base Loss-3-6 | 1.0578 | 1.4757 | 182.3567 | 8.6413 |
| | Base Loss-3-7 | 1.0761 | 1.723 | 203.2225 | 8.2649 |
| Label Length (7 parts) | Label Length-1 | 1.0724 | 1.7383 | 248.3027 | 8.7743 |
| | Label Length-2 | 1.0723 | 1.611 | 246.8519 | 9.2012 |
| | Label Length-3 | 1.0782 | 1.4727 | 280.2898 | 9.664 |
| | Label Length-4 | 1.0595 | 1.3929 | 230.8662 | 10.0671 |
| | Label Length-5 | 1.0644 | 1.3388 | 195.1638 | 10.5148 |
| | Label Length-6 | 1.0681 | 1.2982 | 172.0139 | 11.3054 |
| | Label Length-7 | 1.0763 | 1.2806 | 150.8342 | 9.684 |
| Label Length (7 parts, 1 with ratio 2) | Label Length-2-1 | 1.0678 | 1.3328 | 161.7013 | 8.8855 |
| | Label Length-2-2 | 1.0668 | 1.2999 | 165.7366 | 9.1699 |
| | Label Length-2-3 | 1.0625 | 1.2699 | 184.6405 | 9.4078 |
| | Label Length-2-4 | 1.0686 | 1.2636 | 143.1687 | 9.6699 |
| | Label Length-2-5 | 1.0699 | 1.2598 | 140.6587 | 9.8929 |
| | Label Length-2-6 | 1.0738 | 1.2586 | 139.0638 | 10.146 |
| | Label Length-2-7 | 1.0657 | 1.2582 | 137.4472 | 10.5884 |
| Label Frequency (7 parts) | Label Frequency-1 | 1.1006 | 1.315 | 2226.8591 | 6.454 |
| | Label Frequency-2 | 1.0873 | 1.3696 | 1246.7653 | 6.988 |
| | Label Frequency-3 | 1.085 | 1.3497 | 864.0648 | 7.4164 |
| | Label Frequency-4 | 1.0776 | 1.3177 | 653.4374 | 7.8341 |
| | Label Frequency-5 | 1.0652 | 1.2885 | 544.0231 | 8.1573 |
| | Label Frequency-6 | 1.0673 | 1.266 | 444.3141 | 8.4093 |
| | Label Frequency-7 | 1.0744 | 1.2531 | 387.8805 | 8.636 |

Table 5: Impact of different variables on model performance

### G.3 ANALYSIS OF THE SEMANTIC DISTRIBUTION

Figure 6 presents the semantic spatial distribution of textual data within our dataset. The visualization is generated using t-SNE for dimensionality reduction, followed by DBSCAN clustering to reveal underlying structures among different semantic categories.

From this figure, we can observe several key insights: **(1) Cluster Density and Distribution** - The **dense clusters** indicate high semantic similarity among certain categories, such as: - **Mathematics, Data Science, and STEM Knowledge** (blue, purple) forming compact groups. - **Programming and Software Development** (deep blue) forming a distinct region. - **Legal and Security Knowledge** (red, pink) clustering tightly, indicating semantic coherence. - **Scattered regions** suggest diverse text distributions, such as: - Creative Writing and Social Media (red, brown) overlapping with multiple categories. - Task Management and Counseling spread across different areas.

**(2) Cross-Category Relationships** - Some categories show semantic overlap, suggesting shared contextual usage: - Data Science, Mathematics, and STEM Knowledge exhibit proximity in the space. - Legal, Political, and Security Knowledge share common regions due to regulatory and strategic text overlaps. - Humanities, History, and Philosophy form a loose group with adjacent clusters.

**(3) Implications for Dataset Composition** - The density variation highlights differences in category representation within the dataset. - Sparse clusters may indicate underrepresented categories, suggesting the need for data augmentation. - Overlapping regions suggest semantic drift, which should be considered in downstream NLP applications.

