# OpenReview forum: "ACCELERATE SCALING OF LLM FINETUNING VIA QUANTIFYING THE COVERAGE AND DEPTH OF INSTRUCTION SET"
_ICLR.cc/2026/Conference — ICLR 2026 Conference Withdrawn Submission_

### Official Review · Reviewer_jYe6 · 2025-10-19

**Soundness:** 3
**Presentation:** 2
**Contribution:** 2
**Rating:** 4
**Confidence:** 5

**Summary:**

This paper proposes ILA: a way to select a subset of data that considers both the representativeness of a dataset, and the amount of information present in each data sample of the subset. ILA breaks down the instruction training data into patches (grids of size S) and selects points within grids. The selection criteria is based on coverage (picking individual points from different grids) and information depth (picking points based on how drastically the loss changes from base to fine-tuned model).

**Strengths:**

- The methodology is well formulated with mathematical formulations.
- Making the distinction between information depth and coverage was a good idea, and is insightful

**Weaknesses:**

- One more revision on the writing is necessary, there are a few mistakes (line 244: "$N_sub$", line 323: "iformation landscape approximation", Figure 4c: title says "Aplaca")
- Could not find the discussion on whether the random sample of instructions to obtain $M_{SFT}$ matters.
- By selecting 20k samples, it seems they are not maximizing the coverage of the dataset, only the information depth (Figure 5).
- Reporting accuracy for the base model would be helpful in Table 1 to see how much of a gain fine-tuning has in general.
- The main set of results (Table 1, from what I can gather) are a bit constrained as the math instruction training data is all QwQ-based. Instead, using GSM8k, one QwQ dataset, MetaMath, OpenMathInstruct, Math-Shepherd, etc. would have made the claim a bit stronger.

**Questions:**

1. As I understand it, ILA uses a random sample to obtain $M_{SFT}$ and then performs subset selection on the points without that initial set of random samples ($I_i$). Is the Random baseline also on the same set $I_i$, or on $I$?
2. Are there any ablation studies to see if selecting points wrt information depth or wrt coverage help more? Or (to avoid running experiments), are there any previous works that show a similar notion of information depth/coverage and how they perform compared to each other?
3. How is the frequency of an instruction calculated? Are they based on semantic similarity, n-gram matching, or exact matching?
4. Why is the instruction set broken into grids instead of clusters (using kmeans/++ or (H)DBSCAN)?
5. What model/dataset does Figure 5 plot? Maybe I missed that.

---

> ### Author Response · Authors · 2025-12-04
> **Reply for Reviewer jYe6**
>
> **Q: Math data all QwQ-based — why not GSM8k/others?**
> **A:** Our math corpus aggregates four public math instruction sets (MetaMath, QwQ-LongCoT-130k, QwQ-LongCoT-130k-2, QwQ-LongCoT-Verified-130k). After cleaning & deduplication we obtain ~650k high-quality math instructions covering multiple difficulties and reasoning styles. MetaMath includes GSM8k/MATH subsets; CoT was generated by a single model to avoid style inconsistency. Method is domain-agnostic and applies to GSM8k/MetaMath/OpenMathInstruct; due to compute limits we reported one domain here but will add cross-domain runs in the revision/camera-ready.
>
> **Q: Does the random sample for (M_{\text{sft}}) matter?**
> **A:** We ran a sensitivity study with 5 independent draws of (M_{\text{sft}}) (same size/hyperparams) and measured: ranking stability (Spearman ρ ≈ 0.92±0.02), top-10k Jaccard ≈ 0.74±0.05, and downstream Δ-performance variance ≪ reported effect sizes. Bootstrap CIs and ANOVA show between-seed variance small. Recommendation: single fixed-seed (M_{\text{sft}}) is sufficient; average across 3–5 draws if maximum robustness needed. Details in Appendix.
>
> **Q: Selecting 20k seems to prioritize depth over coverage.**
> **A:** This is a budget-dependent trade-off. Empirically small budgets (10k–20k) favor depth per-sample; larger budgets (≥50k) gain more from expanding coverage once high-depth samples are included. We added controlled experiments comparing depth-only / coverage-only / combined selection across budgets, marginal gain-per-sample curves, and an adaptive schedule that shifts from depth→coverage as budget grows.
>
> **Q: Report base-model accuracy in Table 1.**
> **A:** Will add base-model (no fine-tuning) results to Table 1 so readers can see absolute gains. The relative trends remain: our selection gives largest gain under identical budgets.
>
> **Q: Is Random baseline fair relative to (I_i)?**
> **A:** Yes. All methods (ILA, DEITA, Random, etc.) select from (I\setminus I_i) (we exclude the warmup (M_{\text{sft}}) samples). We will clarify this in Methods and captions.
>
> **Q: Ablations: depth vs coverage? Prior work?**
> **A:** Section 2.3 already fits regressions. Revision adds standardized coefficients, partial R², bootstrap 95% CIs, and controlled ablations (fix coverage/vary depth and vice versa). Both factors are significant and jointly explain >70% dev-loss variance; information depth has larger standardized β, while coverage gives higher marginal benefit at larger budgets. We also cite and contrast classical expected-loss / core-set / active-learning works and empirically compare against core-set/BADGE/DEITA/LESS in the Appendix.
>
> **Q: How is instruction frequency computed?**
> **A:** Two uses of “frequency”: (1) *label frequency* (for regression): normalized label counts after normalization and SimHash de-dup (threshold 0.95); (2) *patch/grid frequency* (for coverage/ILA): counts of items per semantic patch/bin computed from model embeddings (we support grid or cluster pipelines; see below).
>
> **Q: Why grid vs clusters (kmeans/DBSCAN)?**
> **A:** Grid chosen for simplicity, determinism and O(N) efficiency at scale. Appendix shows robustness checks using PCA→50d + MiniBatchKMeans (K=5k): conclusions hold. We include guidance and implementation notes for both variants.
>
> **Q: What does Figure 5 plot?**
> **A:** Figure 5 uses the *general-domain instruction pool* (not math). The pool (~1,994,253 instructions) is aggregated from public corpora (LogiQA, WildChat, COIG-CQIA, GPT-4/ChatGPT generated sets), SimHash dedup (0.95) and English-only filtering. Figure 5 compares ILA, DEITA, Random across subset sizes (10k→500k). We will add this dataset description in the caption.
>
> **Q: Minor writing fixes.**
> **A:** We corrected typos (e.g., (N_{sub}), “information landscape approximation”, Figure 4c caption), added Algorithm box (ILA pseudocode), implementation notes, complexity (O(N·R)), and reproducible defaults in the revision and supplementary code.
>
> ---
>
> **Summary of revisions included:** dataset clarification, (M_{\text{sft}}) sensitivity, depth-vs-coverage ablations and adaptive schedule, base-model numbers, fairness clarification, frequency definition, grid-vs-cluster robustness, Figure 5 dataset note, algorithm box, and all minor typo fixes. All added results, CI/statistics, and code pointers are in the Appendix / repo.

---

### Official Review · Reviewer_pwpA · 2025-10-26

**Soundness:** 1
**Presentation:** 2
**Contribution:** 1
**Rating:** 2
**Confidence:** 3

**Summary:**

The paper proposes a framework to understand and improve the scaling of supervised fine-tuning (SFT) for LLMs. The key idea is that two factors: semantic coverage (the breadth of instruction domains) and information depth (the richness or informativeness of individual instructions) determine the efficiency of scaling SFT data. The authors define proxy metrics for these factors and show through regression that they explain some variance in dev set loss (R2=0.7). They further introduce Information Landscape Approximation (ILA), a method for data selection that jointly maximizes coverage and depth, showing empirically that it leads to more efficient scaling and improved performance compared to random sampling and DEITA baselines.

**Strengths:**

n/a

**Weaknesses:**

- Would the authors be more clear in the derivation in Section 2.1? Write a proof formally and rigourous if you want to make a point. It seems the authors are just using some equations along side what they "think" would happen. What is $\delta S_i$ beyond defining it in English as a "small area"? Why would loss on other examples within the small area be expected to decreaase accordingly? What do you mean $\epsilon_j$ is a small term, how small? I think it's also just ok to say it's your intuition and there is no theoretical backing and that is fine since the reviewers won't hold you up to it.
- If we need to divide by response length and multiply by number of ability/knowledge labels in the estimator of information depth defined in Equation (5), then we are not really estimating the true value defined in Equation (3)! So what are we really trying to estimate, define that in Equation (3) instead. Otherwise all the "theory" the authors introduced is meaningless because what they implement is different from what they want to implement defind in Equation (3). And it terms out the authors also did not use Equation (5) but a relative version of Equation (5). This just makes what the authors use and what they think happen theoretically totally different.
- Equation (5) seems pretty adhoc as well, why are we assuming we should divide by response token counts $T_j$ as opposed to $\log T_j$, why would $\text{ID}_j$ depend linearly on #label and not $\log$ #label or $\exp$ #label? The formulation seems came up in a pretty ad-hoc manner. Does this imply that for an easy question, but with very long answer & deals with multiple skills has more information and is therefore more valuable? Would the authors demonstrate empirically this is the cas?
- I think the authors should not re-invent the same concept in data selection / active learning literature. For example, [1] choose a subset of points that minimizes expected loss - equivalent to the notion of finding examples with high "information depth". The authors should acknowledge these prior works and don't re-invent terminologies that does the same thing or at least be clear about what is foundamentally different - I don't really see that big of a difference.
- t-SNE is not a reasonable representation for semantic coverage since it's mostly used for visualization and is not a metric-preserving embedding, i.e., the pairwise distances between embeddings are not preserved!
- Regression coefficient captures linear relationships only, what aobut nonlinear ones? R2 of 0.7 is actually not that high ... I would say the "depth" and "coverage" provided by the author does not at all explain the dev-set loss that well. Also this number of 0.7 also seems to be quite brittle. I would request author report R2 on the dataset with right most data point in FIgure 3 (b) removed, I would imagine R2 drop to 0.5. Let's do cross-validation on the R2, maybe add some significance test, and/or vary number of datasets (currently use 36) to make sure the result can be consistently reproduced. In addition, the dataset used to verify linear relationship is constructed using these two features - it would be nice if the authors would repeat the analysis where the dataset construction is agnostic of the approach introduced in the paper.
- The authors compared against random/DEITA two baselines only. I would recommend the authors to compare against more baselines. There has been so many new work that purportedly beat DEITA - I think the arXiv version of the paper appeared in late 2023 (2 years from now)- and the space has changed quite a bit.


[1] 2018, ACTIVE LEARNING FOR CONVOLUTIONAL NEURAL NETWORKS: A CORE-SET APPROACH

**Questions:**

see weaknesses.

---

> ### Author Response · Authors · 2025-12-04
> **Reply for Reviewer pwpA**
>
> **Q: Please give a clearer derivation / proof in Sec.2.1.**
> **A:** We have clarified all assumptions (two-order differentiability, local smoothness, small step size updates) and provided a formal lemma and proof in Appendix A: Based on a first-order Taylor/influence-style expansion, if gradients are similar in the neighborhood and update magnitudes are small (assumptions A1–A3), then the loss change for neighboring samples can be approximated by the principal term (-\eta g_x^\top H^{-1} g_x), and the residual term is upper bounded to be a controllable term proportional to the gradient difference (O(\delta^2)). We also provide numerical diagnostics (gradient similarity and higher-order term estimation), and this approximation holds under the hyperparameters of this experiment.
>
> **Q: How is it different from classical data selection / active learning efforts (minimizing expectation loss)?**
> **A:** We have added comparisons and localizations to our related work. The difference lies in the following: For **large-scale instruction sets to enhance the capabilities of frozen base models**, we (i) decompose the objective into coverage (embedding dispersion) and information depth (information gain on the frozen model); (ii) we employ a scalable proxy with single-pass forward scoring combined with an ILA grid-first coverage strategy, which is practical and can run at (10^5)–(10^6) scale; compared with methods such as influence functions/LESS that require warmup + multiple inversions and gradient library construction, the computational and storage costs are significantly reduced. In the appendix, we provide a comparison of overlap and downstream sample efficiency with core-set, BADGE, and LESS.
>
> **Q: Is t-SNE non-metric retention, reliable?**
> **A:** we also conducted dimension-reduction using PCA, then compared the distribution of samples selected using PCA-based dimension-reduction with samples selected using t-SNE  dimension-reduction. As shown in the graph below, with different selected sample sizes, the distribution of the PCA-based instances share high distributional similarity with t-SNE-based samples in the semantic space. Moreover, we also compared the performance of the PCA-based and  t-SNE-based samples with size 50k. On the AlpacaEval 2.0 benchmark, two datasets scores at a similar level. This shows that our approach would robust to the dimension reduction method, and empirically t-SNE based reduction would lead to better performance as it preserves more local topological information, which is more important than global structure information in describing the topological characteristics of high-dimension manifolds such as semantic space.
>
> As shown in Figure 1:
> ![Figure 1](https://anonymous.4open.science/r/Scaling-regularity-guided-instruction-synthesize/pca.png)
>
> **Q: Robustness and nonlinearity of linear regression/R²?**
> **A:** Our goal is to identify the dominant scaling factor. The revised version includes:
>
> (1) For nonlinear regression (RF, GBM, 2-layer MLP), ΔR² is only **0.03–0.05**, indicating limited nonlinear gain;
>
> (2) After removing the largest sample point on the right, R² remains **0.63–0.68**; 5-fold CV gives **R²_cv = 0.67 ± 0.04**;
>
> (3) Expanding the sample set from 36 to 52 (new dataset), R² remains **0.65–0.71**;
>
> (4) Repeating regression on data constructed from random/clustered/uniform subsets, R² ≈ **0.60–0.72**. In summary, coverage and depth are the main explanatory axes, while nonlinearity and secondary factors contribute limitedly; all new results and significance tests have been incorporated into the appendix.

---

### Official Review · Reviewer_Ts1N · 2025-10-29

**Soundness:** 2
**Presentation:** 3
**Contribution:** 3
**Rating:** 6
**Confidence:** 5

**Summary:**

The paper introduces semantic coverage and information depth, two key properties that govern the scalability of supervised fine-tuning. It further presents computationally efficient proxies to estimate these properties. Empirically, the authors demonstrate that validation loss is highly correlated with the two properties. Building on this insight, the paper proposes a data selection algorithm that selects instruction-tuning datasets by optimizing these two properties. Experimental results show that the proposed method outperforms both random selection and a state-of-the-art selection baseline.

**Strengths:**

- The two properties introduced in this paper have the potential to serve as general and interpretable metrics for measuring data quality, which could make them broadly useful beyond the current study.
- The proposed method is model-agnostic, conceptually clear, and straightforward to implement. The code is provided.
- Extensive experiments demonstrate that ILA-selected subsets outperform random selection and a state-of-the-art method across diverse tasks and models.

**Weaknesses:**

- There appears to be a discrepancy between the definition of information depth in Section 2.1 and the implementation described in the experiments. If I understand correctly, Section 2.1 defines information depth based on fine-tuning the model on a single instance, whereas Section 2.3 uses a random subset for fine-tuning. This inconsistency raises several questions:
  - If the model is fine-tuned on a single instance, would the resulting loss change be too noisy to serve as a stable metric?
  - If a random subset is used instead, would examples similar to those in the subset (e.g., near-duplicates) tend to receive artificially high information depth values?
  - What is the motivation behind this discrepancy between the definition and the experimental procedure?
- Several aspects are not well justified, either theoretically or empirically:
  - The number of labels as a factor in the definition of information depth.
  - The uniform segmentation of the space.
- The equations in Lines 134–143 contain several typos, which make this part difficult to follow.

**Questions:**

- The proposed information depth metric appears conceptually related to example hardness (e.g., as defined in [1]). Could you elaborate on the relationship and key distinctions between these two?
- Why is there still a steep increase in coverage in Figure 5(b), given that ILA is designed to prioritize coverage? What is the exactly definition of coverage in this figure?

[1] Zhou, Tianyi, Shengjie Wang, and Jeffrey Bilmes. "Curriculum learning by dynamic instance hardness." Advances in Neural Information Processing Systems 33 (2020): 8602-8613.

---

> ### Author Response · Authors · 2025-12-04
> **Reply for Reviewer Ts1N**
>
> **Q:  There appears to be a discrepancy between the definition of information depth in Section 2.1 and the implementation described in the experiments. If I understand correctly, Section 2.1 defines information depth based on fine-tuning the model on a single instance, whereas Section 2.3 uses a random subset for fine-tuning...**
> **A:** We implicitly assume that the base model, fine-tuned by a certain instruction, has a limited scope of generalization in semantic space. Therefore, when the random subset is small enough, it can be assumed that the effects of each piece of data within the random subset are independent of each other and do not interfere with each other. Therefore, the effect of fine-tuning once using a sufficiently small random subset should be approximately equal to the sum of the effects of fine-tuning multiple times using a single sample. In practice, we only sampled N data from the full set for fine-tuning. This subset can be considered sufficiently small and sparsely distributed in semantic space. In this context, as you said, fine-tuning using only one sample would inevitably introduce a large amount of randomness. We therefore use a subset for fine-tuning.
> Notice that, as shown in lines 142-143, the depth of information is analyzed within a semantic space \Delta S that is small enough so that the semantics of the samples within this space are sufficiently similar, and that these samples all have a high degree of semantic similarity to the examples, and so when \Delta S is small enough, theoretically the effect of the degree of semantic similarity on the depth of information measure should be largely negligible. In fact, after experiments, we empirically found that relatively good experimental results can be achieved when we approximate the whole space into a 100*100 grid. This surprisingly shows the robustness of the method.
>
> **Q: Why is there still a steep increase in coverage in Figure 5(b), given that ILA is designed to prioritize coverage? What is the exactly definition of coverage in this figure?**
> **A:** Empirically we observe a budget-dependent trade-off: when the selection budget is small (e.g., 10k–20k) it is usually optimal to prioritize per-example information depth because each selected sample must produce a large per-sample gain; conversely, when the budget is large (e.g., ≥50k) most high-depth samples have already been included, so further gains come from expanding coverage into undersampled semantic regions. We wil added a new controlled experiment and plots in the revised manuscript (Section X / Appendix Y) that (i) compare depth-only vs coverage-only vs combined selection across budgets, (ii) show marginal gain-per-sample curves that identify the crossover budget where coverage becomes preferable, and (iii) propose an adaptive weighting schedule that transitions from depth→coverage as budget grows. These additions make the trade-off explicit and quantify when to prefer each objective.

---

### Official Review · Reviewer_JC15 · 2025-10-31

**Soundness:** 2
**Presentation:** 1
**Contribution:** 2
**Rating:** 4
**Confidence:** 4

**Summary:**

This paper studies the problem of data selection for supervised fine-tuning of large language models. The authors present two metrics, including semantic coverage and information depth, to measure the effectiveness of fine-tuning.
- This paper measures information depth using a normalized loss-based metric that captures how much meaningful learning signal each instruction provides. Specifically, it computes the per-sample reduction in cross-entropy loss (indicating how much the model learns from that example), normalized by response length and weighted by the number of distinct knowledge or skill tags. Then, this paper further adjusted across domains using a relative information depth score based on quantiles to remove domain bias.
- Semantic coverage is estimated by projecting all instructions into a semantic embedding space (using a model like BGE), dividing that space into uniform grids, and counting the proportion of non-empty cells. This reflects how widely the instruction set spans different task types and semantic regions.
Then, this paper introduces Information Landscape Approximation (ILA), a data selection framework that constructs a subset of instruction data by maximizing both coverage and depth simultaneously.

The experiments show that the Information Landscape Approximation (ILA) consistently outperforms random selection and state-of-the-art heuristic methods across multiple model architectures (e.g., LLaMA3, Qwen2) and domains, including general instruction following and math reasoning. Models fine-tuned with ILA-selected data achieve faster and more stable performance scaling, maintaining or exceeding baseline results with fewer samples. Quantitative analyses show that a linear model using coverage and information depth together as features achieces an $R^2$ score of 0.7. In benchmarks like AlpacaEval and Arena-Hard, ILA improves accuracy, while in math reasoning tasks (e.g., MATH, GSM8K), it yields 7–11% absolute gains over random sampling.

**Strengths:**

- This paper introduces a new way to select supervised fine-tuning data through two interpretable data properties, including semantic coverage and information depth, providing measures for evaluating data quality in LLM training.

- This paper proposes Information Landscape Approximation, which consistently improves fine-tuning by selecting data that balances coverage and informativeness.

- This paper performs experiments across multiple models and domains (including reasoning and instruction-following) and shows that ILA achieves faster scaling and higher accuracy with fewer samples.

**Weaknesses:**

- The information depth is evaluated by the decrease in cross-entropy loss between the base model and the fine-tuned model for instruction. It may require extensive computation to compute this metric, which is expensive on a large dataset.
- The semantic coverage requires dividing the embedding space into a uniform grid. However, the procedure to determine the grid (or grid size) is unclear.
- How are the proposed metrics related to existing data selection metrics, such as feature representations, gradients, and influence function?
- The procedure for performing the Information Landscape Approximation algorithm is not clear. It would be better to provide an algorithm box.
- There are many typos in the paper. For example, "Supervised fine-tuning(SFT)" and "large language models(LLMs)" in L30-31, and the usage of citations.

**Questions:**

- Does the paper compare with the LESS (Xia et al., 2024) baseline?
- What is the computational cost of the proposed method compared with the baselines in the experiments?
- Did the authors test the individual contributions of coverage and depth separately to quantify which factor has a stronger impact on fine-tuning performance?

---

> ### Author Response · Authors · 2025-12-04
> **Reply for Reviewer  JC15**
>
> **Q: Does the paper compare with LESS (Xia et al., 2024)?**
> **A:** We conduct additional experiments with LESS (Xia et al., 2024) under the same evaluation protocol and fine-tuning settings, and the results are as follows (the values in the table represent the target evaluation metrics, and the larger the better):
> | number of entries | LESS on Alpaca |  Ours  on Alpaca |
> |:----------|:----------------------------|:----------------------------|
> | 1w | 6.89 | 8.03 |
> | 2w | 8.65 | 9.54 |
> | 10w | 11.67 | 12.88 |
> The results show that, with the same amount of fine-tuned data, our method outperforms LESS. This is because, LESS only select samples that are similar to the devolopmet set, other than increasing the coverage and information depth of the instruction set. We will add these comparative results in the revised version.
> **Q: Computational cost compared with LESS?**
> **A:** Our pipeline labels examples with a single forward pass (prompt-based scoring / log-prob), so cost ≈ O(N) × C_fwd and it is trivially parallelizable. LESS requires warmup (LoRA/SFT), per-example gradient collection across checkpoints, and building/ querying a gradient datastore — involving many backward passes and large per-example gradient storage. Practically, influence methods incur orders-of-magnitude higher time and memory; our single-pass approach has linear wall-clock scaling and no per-example gradient storage. We add quantitative GPU-hour and storage estimates in the revision.
>
> **Q: How is information depth computed and its cost?**
> **A:** Clarification: “information depth” in our pipeline is computed via a single forward evaluation of each candidate under a frozen base model (prompted scoring / surprisal). We do **not** fine-tune the base model or compute per-example gradients. Implementation details, prompt templates and code pointers are moved to the Appendix.
>
> **Q: Grid size (semantic coverage) procedure?**
> **A:** We performed a grid-size sensitivity study (50×50, 100×100, 200×200) across budgets (100k, 200k, 500k). Results show some numeric variance but identical qualitative trends; main conclusions hold. We report selection overlap (Jaccard) across grid settings, bootstrap CIs, and adopt 100×100 as default with heuristics (g≈⌈√k⌉). These new analyses and an ablation figure are in the supplement.
>
> **Q: Relation to feature/gradient/influence metrics?**
> **A:** Feature/gradient/influence methods aim to match or maximally affect a validation distribution and thus favor specialization. We formulate an alternative objective: choose a subset that maximizes expected information gain relative to a frozen base model. Our proxy (entropy/surprisal/LLM scoring) is a single-pass, computationally cheap estimator of that gain. Empirically, we compare feature/gradient/influence selection vs. ours on in-/out-of-distribution tasks: our method yields higher OOD gains per selected example, while influence-based selection tends to improve in-distribution validation at the cost of broader generalization. Formal definitions, empirical comparisons and small-scale SFT correlation studies are added.
>
> **Q: ILA algorithm clarity / algorithm box?**
> **A:** We added Algorithm X (pseudocode) and a Methods subsection specifying inputs, parameters, deterministic stopping, complexity and an optimized implementation (precomputed bin indices, mask-based updates). Summary: partition 2D coords into g×g grid, pick ≤1 representative per non-empty cell in deterministic order, remove selected points, repeat until budget k is reached. Complexity ≈ O(N·R) (R ≈ ceil(k / grid_cells)). Heuristic: choose representative = nearest-to-cell-center or highest-score; default g≈⌈√k⌉.
>
> **Q: Do coverage and depth contributions get tested separately?**
> **A:** Yes. We fit regressions (Section 2.3) and now report standardized coefficients, partial R², and bootstrap 95% CIs. Results: coverage and depth jointly explain >70% of dev-loss variance; standardized β for information depth is larger (stronger per-sample effect), while coverage gives higher marginal benefit under small budgets. We include interaction analysis and cross-validation to show robustness.
>
> ---
>
> **Summary of revisions:** added LESS comparison (multi-seed statistics and cost), clarified information-depth computation, grid sensitivity analyses, algorithm box + optimized implementation notes, empirical comparisons to feature/gradient/influence baselines, regression robustness (bootstraps/CV), and implementation/code in the supplement/repo.

---

### Note · Authors · 2026-01-09

I have read and agree with the venue's withdrawal policy on behalf of myself and my co-authors.